# INTERPRETABLE AND PEDAGOGICAL EXAMPLES

**Smitha Milli*** **& Pieter Abbeel**
University of California, Berkeley
{smilli,pabbeel}@cs.berkeley.edu

**Igor Mordatch**
OpenAI
{mordatch}@openai.com

## ABSTRACT

Teachers intentionally pick the most informative examples to show their students. However, if the teacher and student are neural networks, the examples that the teacher network learns to give, although effective at teaching the student, are typically uninterpretable. We show that training the student and teacher iteratively, rather than jointly, can produce interpretable teaching strategies. We evaluate interpretability by (1) measuring the similarity of the teacher's emergent strategies to intuitive strategies in each domain and (2) conducting human experiments to evaluate how effective the teacher's strategies are at teaching humans. We show that the teacher network learns to select or generate interpretable, pedagogical examples to teach rule-based, probabilistic, boolean, and hierarchical concepts.

## 1 INTRODUCTION

Human teachers give informative examples to help their students learn concepts faster and more accurately (Shafto et al., 2014; Shafto & Goodman, 2008; Buchsbaum et al., 2011). For example, suppose a teacher is trying to teach different types of animals to a student. To teach what a "dog" is they would not show the student only images of dalmatians. Instead, they would show different types of dogs, so the student generalizes the word "dog" to all types of dogs, rather than merely dalmatians.

Teaching through examples can be seen as a form of communication between a teacher and a student. Recent work on learning emergent communication protocols in deep-learning based agents has been successful at solving a variety of tasks (Foerster et al., 2016; Sukhbaatar et al., 2016; Mordatch & Abbeel, 2017; Das et al., 2017; Lazaridou et al., 2016). Unfortunately, the protocols learned by the agents are usually uninterpretable to humans (Kottur et al., 2017), and thus at the moment have limited potential for communication with humans.

We hypothesize that one reason the emergent protocols are uninterpretable is because the agents are typically optimized jointly. Consider how this would play out with a teacher network $\mathbf{T}$ that selects or generates examples to give to a student network $\mathbf{S}$. If $\mathbf{T}$ and $\mathbf{S}$ are optimized jointly, then $\mathbf{T}$ and $\mathbf{S}$ essentially become an encoder and decoder that can learn any arbitrary encoding. $\mathbf{T}$ could encode "dog" through a picture of a giraffe and encode "siamese cat" through a picture of a hippo.

The examples chosen by $\mathbf{T}$, although effective at teaching $\mathbf{S}$, are unintuitive since $\mathbf{S}$ does not learn in the way we expect. On the other hand, picking diverse dog images to communicate the concept of "dog" is an intuitive strategy because it is the effective way to teach given how we implicitly assume a student would interpret the examples. Thus, we believe that $\mathbf{S}$ having an interpretable learning strategy is key to the emergence of an interpretable teaching strategy.

This raises the question of whether there is an alternative to jointly optimizing $\mathbf{T}$ and $\mathbf{S}$, in which $\mathbf{S}$ maintains an interpretable learning strategy, and leads $\mathbf{T}$ to learn an interpretable teaching strategy. We would ideally like such an alternative to be domain-agnostic. Drawing on inspiration from the cognitive science work on rational pedagogy (see Section 2.1), we propose a simple change:

1. Train $\mathbf{S}$ on random examples
2. Train $\mathbf{T}$ to pick examples for this fixed $\mathbf{S}$

In Step 1, $\mathbf{S}$ learns an interpretable strategy that exploits a natural mapping between concepts and examples, which allows $\mathbf{T}$ to learn an interpretable teaching strategy in Step 2.

---

*Work done while the author was an intern at OpenAI.

We evaluate interpretability in two ways:

1. Evaluating how similar **T**'s strategy is to intuitive human-designed strategies (Section 4)

2. Evaluating how effective **T**'s strategy is at teaching humans (Section 5)

We find that, according to these metrics, **T** learns to give interpretable, pedagogical examples to teach rule-based, probabilistic, boolean, and hierarchical concepts.

## 2 RELATED WORK

### 2.1 RATIONAL PEDAGOGY

What does it mean to rationally teach and learn through examples? One suggestion is that a rational teacher chooses the examples that are most likely to make the student infer the correct concept. A rational student can then update their prior belief of the concept given the examples and the fact that the examples were chosen by a cooperative teacher.

Shafto et al formalize this intuition in a recursive Bayesian model of human pedagogical reasoning (Shafto & Goodman, 2008; Shafto et al., 2012; 2014). In their model the probability a teacher selects an example $e$ to teach a concept $c$ is a soft maximization (with parameter $\alpha$) over what the student's posterior probability of $c$ will be. The student can then update their posterior accordingly. This leads to two recursive equations:

$$P_{\text{teacher}}(e|c) \propto (P_{\text{student}}(c|e))^{\alpha} \tag{1}$$
$$P_{\text{student}}(c|e) \propto P_{\text{teacher}}(e|c)P(c) \tag{2}$$

Note that in general there are many possible solutions to this set of dependent equations. A sufficient condition for a unique solution is an initial distribution for $P_{\text{teacher}}(e|c)$. Shafto et al suggest that a natural initial distribution for the teacher is a uniform distribution over examples consistent with the concept. They empirically show that the fixed point that results from this initial distribution matches human teaching strategies.

In our work, we initialize the teacher distribution in the way suggested by Shafto et al. We optimize in two steps: (1) train the student on this initial distribution of examples (2) optimize the teacher for this fixed student. This approach is analogous to doing one iteration of Equation 2 and then one iteration of Equation 1. We find that one iteration is sufficient for producing interpretable strategies.

### 2.2 COMMUNICATION PROTOCOL LEARNING.

Teaching via examples can be seen as communication between a teacher to a student via examples. Much recent work has focused on learning emergent communication protocols in deep-learning based agents (Foerster et al., 2016; Sukhbaatar et al., 2016). However, these emergent protocols tend to be uninterpretable (Kottur et al., 2017). A number of techniques have been suggested to encourage interpretability, such as limiting symbol vocabulary size (Mordatch & Abbeel, 2017), limiting memorization capabilities of the speaker (Kottur et al., 2017), or introducing auxiliary tasks such as image labelling based on supervision data (Lazaridou et al., 2016).

Despite these modifications, the protocols can still be difficult to interpret. Moreover, it is unclear how modifications like limiting vocabulary size apply when communication is in the form of examples because usually examples are already a fixed length (e.g coordinates in a plane) or constrained to be selected from a set of possible examples. So, there must be other reasons that humans come up with interpretable protocols in these settings, but neural networks do not.

We suggest that one reason may be that these communication protocols are typically learned through joint optimization of all agents (Foerster et al., 2016; Sukhbaatar et al., 2016; Mordatch & Abbeel, 2017; Kottur et al., 2017; Lazaridou et al., 2016), and evaluate how changing from a joint optimization to an iterative one can improve interpretability.

---

**Algorithm 1** Joint Optimization

---

**Require:** $p(\mathcal{C})$: distribution over concepts
   **while** not converged **do**
      Sample $c_1, \ldots c_n \sim p(\mathcal{C})$
      **for** each $c_i$ **do**
         Initialize $\hat{c}_{i,0} = 0$
         **for** $k \in \{1, ..., K\}$ **do**
            $e_k = \mathbf{T}(c_i, \hat{c}_{i,k-1} | \theta_\mathbf{T})$
            $\hat{c}_{i,k} = \mathbf{S}(e_k | \theta_\mathbf{S})$
         **end for**
      **end for**
      $\theta_\mathbf{S} = \theta_\mathbf{S} - \frac{1}{n} \nabla_{\theta_\mathbf{S}} \sum_i \sum_k \mathcal{L}(c_i, \hat{c}_{i,k})$
      $\theta_\mathbf{T} = \theta_\mathbf{T} - \frac{1}{n} \nabla_{\theta_\mathbf{T}} \sum_i \sum_k \mathcal{L}(c_i, \hat{c}_{i,k})$
   **end while**

---

## 2.3 INTERPRETABILITY IN MACHINE TEACHING.

One problem studied in the machine teaching literature is finding a student-teacher pair such that the student can learn a set of concepts when given examples from the teacher (Jackson & Tomkins, 1992; Balbach & Zeugmann, 2009). However, it is difficult to formalize this problem in a way that avoids contrived solutions known as "coding tricks." Although the community has not agreed on a single definition of what a coding trick is, it refers to a solution in which the teacher and student simply "collude" on a pre-specified protocol for encoding the concept through examples.

Many additional constraints to the problem have been proposed to try to rule out coding tricks. These additional constraints include requiring the student be able to learn through any superset of the teacher's examples (Goldman & Mathias, 1996), requiring the learned protocols to work for any ordering of the concepts or examples (Zilles et al., 2011), requiring the student to learn all concepts plus their images under primitive recursive operators (Ott & Stephan, 2002), and giving incompatible hypothesis spaces to the student and teacher (Angluin & Kriķis, 1997).

The prior work has mainly been theoretically driven. The papers provide a definition for what it means to avoid collusion and then aim to find student-teacher pairs that provably satisfy the proposed definition. Our work takes a more experimental approach. We provide two criteria for interpretability and then empirically evaluate how modifying the optimization procedure affects these two criteria.

## 3 APPROACH

We consider a set of possible concepts $\mathcal{C}$ and examples $\mathcal{E}$. For example, $\mathcal{C}$ may be different animals like cats, dogs, parrots, etc and $\mathcal{E}$ may be images of those animals. The prior $p(e|c)$ is a distribution over non-pedagogically selected examples of the concept. For example, if $\mathcal{C}$ is the set of all animals, then $p(e|c)$ could be a uniform distribution over images of a given animal.

A student $\mathbf{S} : \mathcal{E} \mapsto \mathcal{C}$ takes in a running sequence of $K$ examples and at each step outputs a guess $\hat{c}$ for the concept the sequence of examples corresponds to. A teacher $\mathbf{T} : \mathcal{C} \times \mathcal{C} \mapsto \mathcal{E}$ takes in the target concept to teach and $\mathbf{S}$'s current guess of the concept and outputs the next example for the student at each step. When the set of examples is continuous $\mathbf{T}$ outputs the examples directly. When $\mathcal{E}$ is discrete we use the Gumbel-Softmax trick (Jang et al., 2016) to have $\mathbf{T}$ generate a sample from $\mathcal{E}$.

The performance of both $\mathbf{S}$ and $\mathbf{T}$ is evaluated by a loss function $\mathcal{L} : \mathcal{C} \times \mathcal{C} \mapsto \mathbb{R}$ that takes in the true concept and $\mathbf{S}$'s output after $K$ examples (although in some tasks we found it useful to sum the losses over all $\mathbf{S}$'s outputs). In our work, both $\mathbf{S}$ and $\mathbf{T}$ are modeled with deep recurrent neural networks parameterized by $\theta_\mathbf{S}$ and $\theta_\mathbf{T}$, respectively. Recurrent memory allows the student and teacher to effectively operate over sequences of examples. $\mathbf{T}$ and $\mathbf{S}$ are illustrated graphically in Figure 1.

In the recent work on learning deep communication protocols, the standard way to optimize $\mathbf{S}$ and $\mathbf{T}$ would be to optimize them jointly, similar to the training procedure of an autoencoder (Algorithm 1). However, joint optimization allows $\mathbf{S}$ and $\mathbf{T}$ to form an arbitrary, uninterpretable encoding of

---

**Algorithm 2** Best Response (BR) Optimization

---

**Require:** $p(\mathcal{C})$: distribution over concepts
   *Train student on random examples:*
   **while** not converged **do**
      Sample $c_1, \ldots c_n \sim p(\mathcal{C})$
      **for** each $c_i$ **do**
         **for** $k \in \{1, ..., K\}$ **do**
            $e_k \sim p(\cdot | c_i)$
            $\hat{c}_{i,k} = \mathbf{S}(e_k | \theta_\mathbf{S})$
         **end for**
      **end for**
      $\theta_\mathbf{S} = \theta_\mathbf{S} - \frac{1}{n} \nabla_{\theta_\mathbf{S}} \sum_i \sum_k \mathcal{L}(c_i, \hat{c}_{i,k})$
   **end while**
   *Train teacher best response to student:*
   **while** not converged **do**
      Sample $c_1, \ldots c_n \sim p(\mathcal{C})$
      **for** each $c_i$ **do**
         Initialize $\hat{c}_{i,0} = 0$
         **for** $k \in \{1, ..., K\}$ **do**
            $e_k = \mathbf{T}(c_i, \hat{c}_{i,k-1} | \theta_\mathbf{T})$
            $\hat{c}_{i,k} = \mathbf{S}(e_k | \theta_\mathbf{S})$
         **end for**
      **end for**
      $\theta_\mathbf{T} = \theta_\mathbf{T} - \frac{1}{n} \nabla_{\theta_\mathbf{T}} \sum_i \sum_k \mathcal{L}(c_i, \hat{c}_{i,k})$
   **end while**

---

the concept via examples. We compare joint optimization to an alternative approach we call a best response (BR) optimization (Algorithm 2), which iteratively trains $\mathbf{S}$ and $\mathbf{T}$ in two steps:

1. Train $\mathbf{S}$ on concept examples $e_1, \ldots e_K \sim p(\cdot | c)$ coming from prior example distribution.
2. Train $\mathbf{T}$ to select or generate examples for the fixed $\mathbf{S}$ from Step 1.

The intuition behind separating the optimization into two steps is that if $\mathbf{S}$ learns an interpretable learning strategy in Step 1, then $\mathbf{T}$ will be forced to learn an interpretable teaching strategy in Step 2. [1] The reason we expect $\mathbf{S}$ to learn an "interpretable" strategy in Step 1 is that it allows $\mathbf{S}$ to learn a strategy that exploits the natural mapping between concepts and examples. For example, suppose the concept space is the set of all rectangles and $p(e|c)$ is a uniform distribution over points within a rectangle (the task in Section 4.1). In Step 1, $\mathbf{S}$ learns to only guess rectangles that contain all the given examples. Because $\mathbf{S}$ expects examples to be within the rectangle, then in Step 2, $\mathbf{T}$ learns to only give examples that are within the rectangle, *without explicitly being constrained to do so*. So, $\mathbf{T}$ learns to picks the most informative examples that are still within the rectangle, which are the corners of the rectangle.

## 4 EXPERIMENTS

The purpose of our experiments is to examine what kind of emergent teaching strategies $\mathbf{T}$ learns and whether or not they are *interpretable*. However, there are many definitions of interpretability in the literature (Doshi-Velez & Kim, 2017; Weller, 2017; Lipton, 2016). Rather than selecting just one, we evaluate interpretability in two ways, hoping that together these evaluations more robustly capture what we mean by interpretability. We evaluate interpretability by:

1. Evaluating how similar $\mathbf{T}$'s strategies are to intuitive human-designed strategies in each task
2. Evaluating the effectiveness of $\mathbf{T}$'s strategy at teaching humans.

---

[1] We also explored doing additional best responses, but this did not increase interpretability compared to just one best response. In addition, we explored optimizing $\mathbf{S}$ and $\mathbf{T}$ jointly after pre-training $\mathbf{S}$ with Step 1, but this did not lead to more interpretable protocols than directly training jointly.

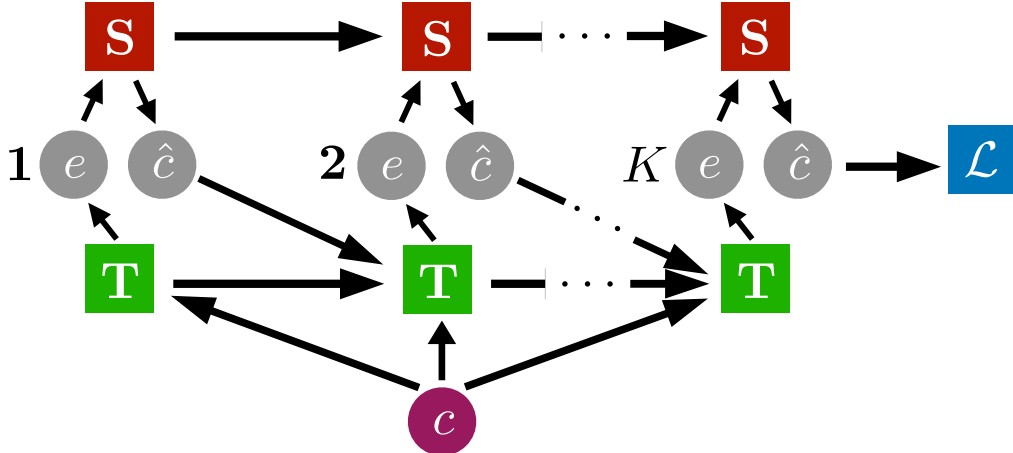

Figure 1: A visualization of the interaction between **T** and **S**. At each step **T** takes in the true concept and **S**'s last estimate of the concept and outputs an example for **S**. Then **S** outputs its new estimate of the concept.

We created a variety of tasks for evaluation that capture a range of different types of concepts (rule-based, probabilistic, boolean, and hierarchical concepts). Below we give a brief description of the tasks and why we chose them. The rest of the section provides further details on the tasks and the first interpretability criteria, while the next section addresses the second interpretability criteria.

**Rule-based concepts.** We first aimed to replicate a common task in the rational pedagogy literature in cognitive science, known as the *rectangle game* (Shafto & Goodman, 2008). In the variant of the rectangle game that we consider, there is a rectangle that is known to the teacher but unknown to the student. The student's goal is to infer the boundary of the rectangle from examples of points within the rectangle. The intuitive strategy that human teachers tend to use is to pick opposite corners of the rectangle (Shafto et al., 2012; 2014). We find that **T** learns to match this strategy.

**Probabilistic concepts.** It is often difficult to define naturally-occurring concepts via rules. For example, it is unclear how to define what a bird is via logical rules. Moreover, some examples of a concept can seem more prototypical than others (e.g sparrow vs peacock) (Rosch & Mervis, 1975), and this is not captured by simply modeling the concept as a set of rules that must be satisfied. An alternative approach models concept learning as estimating the probability density of the concept (Anderson, 1991; Ashby & Alfonso-Reese, 1995; Fried & Holyoak, 1984; Griffiths et al., 2008).

Shafto et al. (2014) investigate teaching and learning unimodal distributions. But often a concept (e.g lamp) can have multiple subtypes (e.g. desk lamp and floor lamp). So, we investigate how **T** teaches a bimodal distribution. The bimodal distribution is parameterized as a mixture of two Gaussian distributions and **S**'s goal is to learn the location of the modes. **T** learns the intuitive strategy of giving examples at the two modes.

**Boolean concepts.** An object can have many properties, but only a few of them may be relevant for deciding whether the object belongs to a concept or not. For example, a circle is a circle whether it has a radius of 5 centimeters or 100 meters. The purpose of this task is to see what strategy **T** learns to quickly teach **S** which properties are relevant to a concept.

The possible examples we consider are images that vary based on four properties: size (small, medium, large), color (red, blue, green), shape (square vs circle), and border (solid vs none). Only one to three of these properties define a concept. For example, if the concept is red circles, then red circles of any size or border fit the concept.

**T** learns the intuitive strategy of picking two examples whose only common properties are the ones required by the concept, allowing **S** to learn that the other properties are not relevant for membership in the concept.

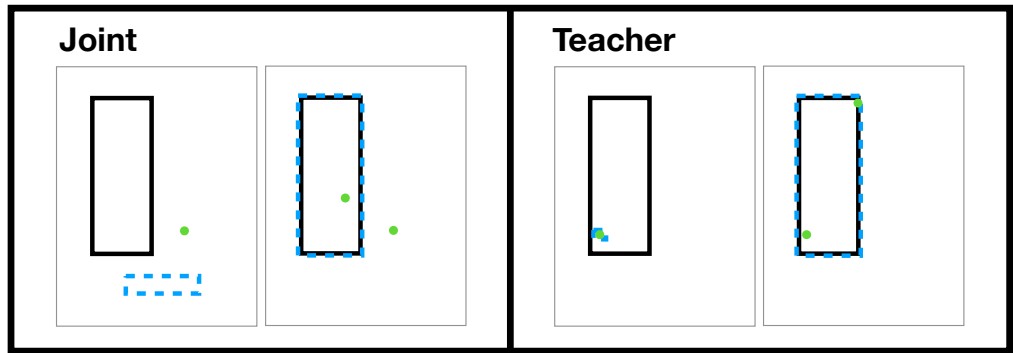

Figure 2: *Rule-based concepts.* The black rectangle is the ground-truth concept and the blue dashed rectangle is student's output after each example. Left: The joint optimization has no clear interpretable strategy. Right: Under BR optimization **T** learns to give opposite corners of the rectangle.

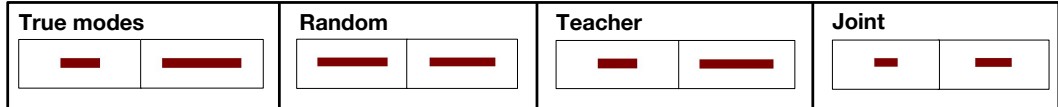

Figure 3: *Probabilistic concepts.* **T** picks examples at different modes more consistently than the random policy, which picks examples near the same mode half of the time. Example are visualized by length of lines.

**Hierarchical concepts.** Human-defined concepts are often hierarchical, e.g. animal taxonomies. Humans are sensitive to taxonomical structure when learning how to generalize to a concept from an example (Xu & Tenenbaum, 2007). The purpose of this task is to test how **T** learns to teach when the concepts form a hierarchical structure. We create hierarchical concepts by pruning subtrees from Imagenet. **T**'s goal is to teach **S** nodes from any level in the hierarchy, but can only give images from leaf nodes. **T** learns the intuitive strategy of picking two examples whose lowest common ancestor is the concept node, allowing **S** to generalize to the correct level in the hierarchy.

### 4.1 RULE-BASED CONCEPTS

A concept (rectangle) is encoded as a length four vector $c \in [-10, 10]^4$ of the minimum x, minimum y, maximum x, and maximum y of the rectangle. $p(e|c)$ is a uniform distribution over points in the rectangle. Examples are two-dimensional vectors that encode the x and y coordinate of a point. The loss between the true concept $c$ and **S**'s output $c'$ is $\mathcal{L}(c, \hat{c}) = ||c - \hat{c}||_2^2$ and is only calculated on **S**'s last output. **S** is first trained against ten examples generated from $p(e|c)$. Then **T** is trained to teach **S** in two examples. **T** generates examples continuously as a two-dimensional vector.

Figure 2 shows an example of **T**'s choices and **S**'s guess of the concept after each example given. Under both BR and joint optimization **S** is able to infer the concept in two examples. However, in joint optimization it is not clear how **T**'s examples relate to the ground-truth rectangle (black) or what policy the student (orange) has for inferring the rectangle. On the other hand, in the BR case **T** outputs points close to opposite corners of the rectangle, and **S** expands its estimate of the rectangle to fit the examples the teacher gives.

Figure 4 measures the distance between the random, best response (teacher), and joint strategy to the intuitive strategy of giving corners averaged over concepts. Specifically, let $e = (e_1, e_2)$ be the two examples given and $S(c)$ be the set of tuples of opposite corners of $c$. The distance measures how close these two examples are to a pair of opposite corners and is defined as $d(e, c) = \min_{s \in S(c)} ||e_1 - s_1||_2 + ||e_2 - s_2||_2$. **T**'s examples are much closer to opposite corners than either the random or joint strategy.

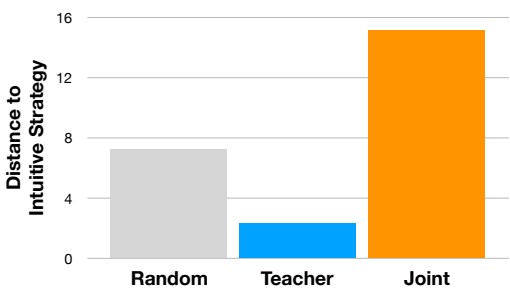
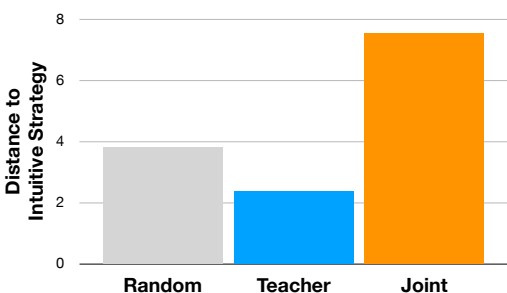

Figure 4: *Rule-based concepts.* **T**'s examples are closer to oppposite corners of the rectangles than randomly generated or jointly trained examples.

Figure 5: *Probabilistic concepts.* **T**'s examples are closer to the two modes than randomly generated or jointly trained examples.

## 4.2 PROBABILISTIC CONCEPTS

A concept is encoded as two-dimensional vector $c = (\mu_1, \mu_2) \in [0, 20]^2$ where $\mu_1$ and $\mu_2$ are the locations of the two modes and $\mu_1 < \mu_2$. $p(e|c) = 0.5\mathcal{N}(\mu_1, 1) + 0.5\mathcal{N}(\mu_2, 1)$ is a mixture of two Gaussians. The loss between the true concept $c$ and **S**'s output $\hat{c}$ is $\mathcal{L}(c, \hat{c}) = ||c - \hat{c}||_2^2$. **S** is first trained against five examples generated from $p(e|c)$. Then **T** is trained to teach **S** in two examples. **T** generates examples continuously as a one-dimensional vector.

**T** learns the intuitive strategy of giving the two modes as the examples. Figure 5 measures the distance to the intuitive strategy by the distance, $||e - c||_2$, between the examples, $e$, and the true modes, $c$. Both $e$ and $c$ are sorted when calculating the distance. **T** learns to match the intuitive strategy better than the random or joint strategy.

Figure 3 shows an example of the choices of the random, teacher, and joint strategy. While the random strategy sometimes picks two examples closer to one mode, **T** is more consistent about picking examples at two of the modes (as indicated by Figure 5). It is unclear how to interpret the choices from the joint strategy.

## 4.3 BOOLEAN CONCEPTS

Examples are images of size 25 x 25 x 3. Concepts are ten-dimensional binary vectors where each dimension represents a possible value of a property (size, color, shape, border). The value of one in the vector indicates that the relevant property (e.g. color) must take on that value (e.g. red) in order to be considered a part of the

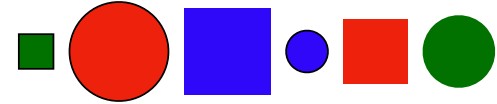

Figure 6: *Boolean concepts.* Possible example images.

concept. $p(e|c)$ is a uniform distribution over positive examples of the concept. The loss between the true concept $c$ and **S**'s output $\hat{c}$ is $\mathcal{L}(c, \hat{c}) = ||c - \hat{c}||_2^2$. **S** is first trained on five examples generated from $p(e|c)$. In both BR and joint optimization, we trained **S** with a curriculum starting with concepts defined by three properties, then two, and then one. **T** is trained to teach **S** with two examples. In this experiment, **T** selects an example from a discrete set of all images. We use the Gumbel-Softmax estimator (Jang et al., 2016) to select discrete examples from final layer of **T** in a differentiable manner.

**T** learns the intuitive strategy of picking two examples whose only common properties are the ones required by the concept, so that **S** can rule out the auxiliary properties. For example, Figure 7 shows **T**'s examples for the concept of red. **T** selects a large red square with no border and then a small red circle with a border. The only property the two shapes have in common is red, so the concept must only consist of red. Indeed, 87% of **T**'s examples only have the required properties in common, compared to 36% of random examples, and 0% of jointly trained examples (Figure 8).

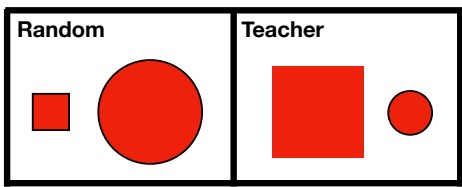

Figure 7: *Boolean concepts.* Examples for the concept "red". Left: The concept "red with border" and "red" are consistent with the random examples. Right: Only the true concept "red" is consistent with **T**'s examples.

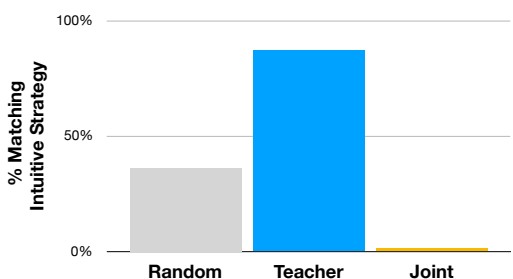

Figure 8: *Boolean concepts.* **T** matches the intuitive strategy 87% of the time, compared to 36% for random, and 0% for joint.

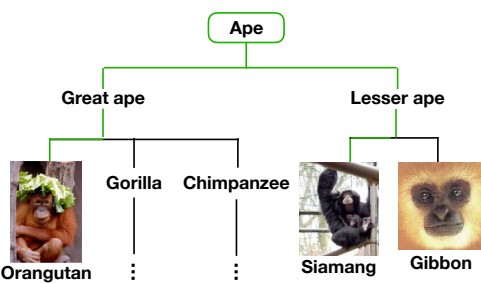

Figure 9: *Hierarchical concepts.* An example subtree. **T**'s strategy is to give two nodes whose lowest common ancestor is the target concept. For example, to teach ape **T** could choose to give an orangutan image and a siamang image.

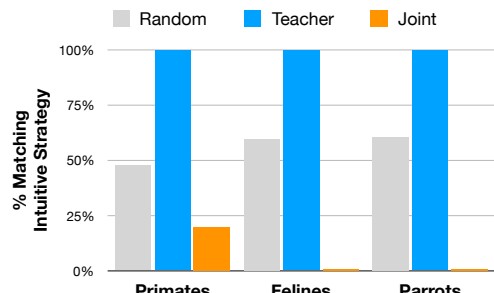

Figure 10: *Hierarchical concepts.* **T** learns to perfectly match the intuitive strategy for hierarchical concepts, but the joint optimization matches the intuitive strategy less than random examples.

## 4.4 HIERARCHICAL CONCEPTS

We create a set of hierarchical concepts by pruning a subtree from Imagenet. Each node in the subtree is a concept and is encoded as a one-hot vector. We randomly select 10 images of each leaf node. The possible examples for a leaf node are any of its ten images. The possible examples for an interior node are images from any of its descendant leaves. For example, in the hierarchy of apes shown in Figure 9, the possible examples for the "lesser apes" concept are images of siamangs or gibbons.

We use a pretrained ResNet-50 model (He et al., 2015) to embed each image into a 2048 length vector. $p(e|c)$ is a uniform distribution over the possible examples for the concept. $\mathcal{L}(c, \hat{c})$ is the softmax cross entropy loss between the true concept $c$ and **S**'s output $\hat{c}$. **S** is first trained on five examples generated from $p(e|c)$. **T** then learns to teach **S** with two examples. As in 4.3, the final layer of **T** uses the Gumbel-Softmax estimator to sample an example image.

**T** learns the intuitive strategy of picking examples from two leaf nodes such that the lowest common ancestor (LCA) of the leaf nodes is the concept node. This strategy encodes the intuition that to teach someone the concept "dog" you wouldn't only show them images of dalmations. Instead you would show examples of different types of dogs, so they generalize to a higher level in the taxonomy. For example, to teach what an ape is **T** could select an image of an orangutan and a siamang because the lowest common ancestor of the two is the ape concept (Figure 9).

Figure 10 shows **T**'s correspondence to the intuitive strategy on the interior nodes of three example subtrees of Imagenet: apes, parrots, and felines. These subtrees have 16, 19, and 57 possible concepts respectively. **T** learns to follow the LCA strategy 100% of the time, whereas the highest the jointly trained strategy ever gets is 20%.

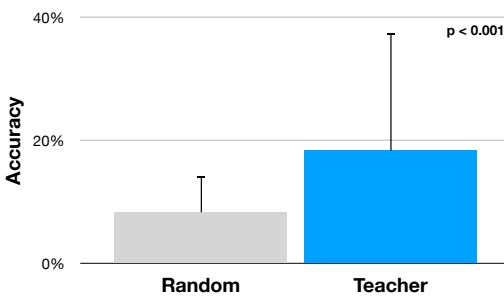 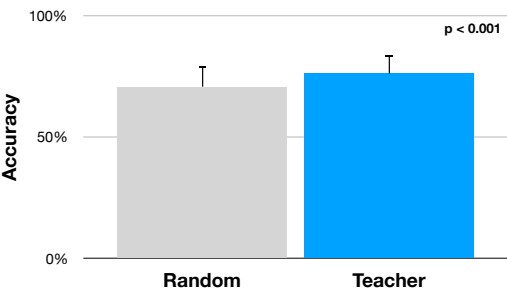

Figure 11: *Probabilistic concepts.* Humans learned the correct distribution over concepts better than humans given random examples.

Figure 12: *Boolean concepts.* Humans learned to classify test images better through examples from **T**.

## 5 TEACHING HUMANS

In the previous section, we evaluated interpretability by measuring how similar **T**'s strategy was to a qualitatively intuitive strategy for each task. In this section, we revisit two of the tasks and provide an additional measure of interpretability by evaluating how effective **T**'s strategy is at teaching humans.

### 5.1 PROBABILISTIC CONCEPTS

We ran experiments to see how well **T** could teach humans the bimodal distributions task from Section 4.2. 60 subjects were recruited on Amazon Mehcanical Turk. They were tested on the ten concepts with modes in $\mathcal{E} = \{4, 8, 12, 16, 20\}$. 30 subjects were shown two examples generated from $p(e|c)$ for each concept and the other 30 subjects were shown two examples generated by **T** for each concept. The subjects were then given five test lines of lengths in $\mathcal{E}$ and asked to rate on a scale of one to five how likely they think the line is a part of the concept. For each concept there were two lines with very high probability of being in the concept and three lines with very low probability of being in the concept. A subject is said to have gotten the concept correct if they gave the high-probability lines a rating greater than three and the low-probability lines a rating less than or equal to three.

The subjects given examples from the teacher had an average accuracy of 18%, compared to 8% with random examples. In addition, the teacher group had a much higher standard deviation than the random group, 19% compared to 6%. The maximum accuracy in the teacher group was 70%, but just 20% in the random group. The difference between groups was highly significant with $p < 0.001$, calculated using a likelihood-ratio test on an ordinary logit model as described in Jaeger (2008).

Although the teacher group did better, neither group had a high mean accuracy. The task is difficult because a subject needs to get the entire distribution correct to be counted as a correct answer. But another possible reason for poor performance is people may have had the wrong hypothesis about the structure of concepts. It seems as though many subjects hypothesized that the structure of the concept space was unimodal, rather than bimodal, thus believing that lines with a length in between the two shown to them were very likely to be a part of the concept. An interesting open research question is how to ensure that the human has the correct model of the concept space.

### 5.2 BOOLEAN CONCEPTS

To evaluate human learning of boolean concepts (the task from Section 4.3), we sampled ten test concepts, five composed of one property and five composed of two properties. We recruited 80 subjects on Amazon Mechanical Turk and showed 40 of them two random positive examples of the ten concepts and the other 40 of them two examples chosen by the teacher. They were then asked to classify four new images as either a part of the concept or not. The four new images always had two positive examples and two negative examples for the concept. As shown in Figure 12, the group that received examples from **T** performed better with a mean accuracy of 76%, compared to a mean accuracy of 71% for those that received random examples. This difference was highly significant with $p < 0.001$, calculated using the same procedure described in Section 5.1 from Jaeger (2008).

## 6 DISCUSSION

What leads the protocols that humans learn to be so different from the protocols that deep learning models learn? One explanation is that humans have limitations that deep learning models do not.

We investigated the impact of one limitation: humans cannot jointly optimize among themselves. We found that switching to an iterative optimization in which (1) the student network is trained against examples coming from a non-pedagogical distribution and then (2) the teacher network is trained against this fixed student leads to more interpretable teaching protocols. The intuition behind the approach is that (1) leads the student to learn an interpretable learning strategy, which then constrains the teacher to learn an interpretable teaching strategy in (2).

But this is just one of many possible limitations. For example, one reason we believe human students did not learn concepts as well as the student network (Section 5) is that humans had a different prior over concepts. In the probabilistic concepts task, humans seemed to believe that the lines came from a unimodal, rather than bimodal, distribution. In the boolean concepts task, humans tended to overemphasize color as a property. It is unrealistic to assume that a teacher and student have a perfectly matching prior over concepts or perfect models of each other. An important open question is which of these limitations are fundamental for the emergence of interpretable teaching protocols.

While we carried out our experiments in the setting of teaching via examples, another direction for future work is investigating how an iterative optimization procedure works in more complex teaching settings (say teaching through demonstrations) and in communication tasks more broadly.

Overall, we hope that our work presents a first step towards understanding the gap between the interpretability of machine agents and human agents.

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
