# OpenReview forum: "Interpretable and Pedagogical Examples"
_ICLR.cc/2018/Conference — Reject_

### Official Review · AnonReviewer2 · 2017-11-21
**This is a compelling paper on an interesting topic: the interpretability of learning strategies.**

**Rating:** 8
**Confidence:** 3

**Review:**

This is a well written paper on a compelling topic: how to train "an automated teacher" to use intuitive strategies  that would also apply to humans.

The introduction is fairly strong, but this reviewer wishes that the authors would have come up with an intuitive example that illustrates why the strategy "1) train S on random exs; 2) train T to pick exs for S" makes sense. Such an example would dramatically improve the paper's readability.

The paper appears to be original, and the related work section is quite extensive.

A second significant improvement would be to add an in-depth  running example in section 3, so that the authors could illustrate why the BR strategy makes sense (Algorithm 2).

---

> ### Author Response · Authors · 2018-01-02
> **Exposition**
>
> Thank you for the kind words overall and the suggestion with regards to the exposition. We realized that although we gave intuition for why the joint strategy would not produce interpretable results in the introduction, we did not give much intuition for why the best response strategy would produce interpretable results. The intuition is essentially that step one allows the student to learn the relation between examples and concepts, and then in step two, the teacher exploits this to give informative examples. Since the teacher is exploiting the relationship between concepts and examples, the examples look interpretable because they are still grounded to the concepts.
>
> For example, for the case with the rectangles, training on random examples in step one allows the student learns that the perimeter of the rectangle must contain all examples given to it. It learns a strategy of guessing approximately the smallest rectangle that will encompass all examples. Because the student expects examples to be within the rectangle, then in step two, the teacher automatically learns to give examples that are within the rectangle, without being constrained to. Thus, the teacher learns to pick the most informative examples that are still within the rectangle, which are the corners of the rectangle.
>
> We have incorporated this extra exposition into the updated version of the paper, in the introduction and Section 3.

---

### Official Review · AnonReviewer3 · 2017-11-27
**A great first-step towards interpretable teaching for deep-learning methods**

**Rating:** 8
**Confidence:** 4

**Review:**

The authors define a novel method for creating a pair of models, a student and a teacher model, that are co-trained in a manner such that the teacher provides useful examples to the student to communicate a concept that is interpretable to people. They do this by adapting a technique from computational cognitive science called rational pedagogy. Rather than jointly optimize the student and teacher (as done previously), they have form a coupled relation between the student and teacher where each is providing a best response to the other. The authors demonstrate that their method provides interpretable samples for teaching in commonly used psychological domains and conduct human experiments to argue it can be used to teach people in a better manner than random teaching.

Understanding how to make complex models interpretable is an extremely important problem in ML for a number of reasons (e.g., AI ethics, explainable AI). The approach proposed by the authors is an excellent first step in this direction, and they provide a convincing argument for why a previous approach (joint optimization) did not work. It is an interesting approach that builds on computational cognitive science research and the authors provide strong evidence their method creates interpretable examples. They second part of their article, where they test the examples created by their models using behavioral experiments was less convincing. This is because they used the wrong statistical tests for analyzing the studies and it is unclear whether their results would stand with proper tests (I hope they will! – it seems clear that random samples will be harder to learn from eventually, but I also hoped there was a stronger baseline.).

For analysis, the authors use t-tests directly on KL-divergence and accuracy scores; however, this is inappropriate (see Jaeger, 2008; Categorical data analysis: Away from ANOVAs (transformation or not) and towards logit mixed models. Journal of Memory and Language, 59(4), 434-446.). This is especially applicable to the accuracy score results and the authors should reanalyze their data following the paper referenced above. With respect to KL-divergence, a G-test can be used (see https://en.wikipedia.org/wiki/G-test#Relation_to_Kullback.E2.80.93Leibler_divergence). I suspect the results will still be meaningful, but the appropriate analysis is essential to be able to interpret the human results.

Also, a related article: One article testing rational pedagogy in more ML contexts and using it to train ML models that is
Ho, M. K., Littman, M., MacGlashan, J., Cushman, F., & Austerweil, J. L. (NIPS 2016). Showing versus Doing. Teaching by Demonstration.

For future work, it would be nice to show that the technique works for finding interpretable examples in more complex deep learning networks, which motivated the current push for explainable AI in the first place.

---

> ### Author Response · Authors · 2018-01-02
> **Analysis of human experiments**
>
> Thank you for the kind words overall and the helpful remarks around the statistical tests for the human experiments. We redid the analysis of the boolean concepts human experiment (the accuracy score) following the methodology in [Jaeger, 2008] and found the result to still be highly significant (p < 0.001).
>
> Our goal in the bimodal concepts human experiment (scored with KL divergence) was to test whether humans given examples from the teacher (Group 1) learn the bimodal distributions better than humans given random examples (Group 2). We thought a natural way to quantify this would be to calculate for each concept the KL divergence between a subject’s estimated distribution and the true ground-truth distribution. We then attempted to see if there was a significant difference between the KL divergences in Group 1 and Group 2 using a t-test.
>
> We did not see how this translates to a G-test because we want to compare the KL divergences between the two groups, rather than testing whether the human distributions are different from the theoretical, true distribution.
>
> As you mentioned, proper statistical tests are crucial to interpret our results. We realized another obstacle to easily interpreting our results is the interpretability of the measure. It is difficult for anyone to interpret what the KL divergence really means in this context.
>
> So, we devised a more direct measure that also lent itself to clear statistical analysis. We emphasize that we did not change any of the data in the experiment, we merely reanalyzed it with a less obfuscated measure than KL divergence.
>
> In the experiment, subjects were shown five test lines for each concept, two of which had a high probability of being in the concept and three of which had a very low probability of being in the concept. They were asked to rate on a scale from 1 to 5 how likely it was that a line was in the concept. Rather, than calculating KL divergence, we calculate an accuracy score where a success is defined as rating the high-probability lines > 3 and the low-probability lines <= 3.
>
> Using an accuracy measure leads to much more interpretable analysis. Here are the statistics for the teacher and random group:
>
> Group 1 (Teacher exs):
> Mean accuracy: 0.183, Min accuracy: 0, Max accuracy: 0.7, Standard deviation: 0.190
>
> Group 2 (Random exs):
> Mean accuracy: 0.083, Min accuracy: 0, Max accuracy: 0.2, Standard deviation: 0.058
>
> Using this measure makes it more clear how well people are actually doing. It is a hard task because they are tested on getting the entire distribution correct. And as you can see, neither group does very well. As mentioned in the paper, we believe a reason for the low accuracy is that humans have a different, incorrect prior over the structure of concepts. In particular, it seems like many believed that the concepts were unimodal distributions, rather than bimodal.
>
> However, the teacher group is better on average (18% versus 8%), and has much more variance, with the best getting 70% of the questions right, whereas the best under random examples is only 20%.
>
> These differences were highly significant (p < 0.001), as calculated using the methodology from [Jaeger, 2008]. We also now share identical analytical methodology across our two experiments.
>
> We have uploaded a revised version of the paper with these modifications in Section 5. Thank you again for the information regarding the statistical tests. It is important to us to ensure that our analysis is solid, and we welcome any further suggestions.

---

### Official Review · AnonReviewer1 · 2017-12-01
**Interesting idea, but not convincing enough**

**Rating:** 4
**Confidence:** 3

**Review:**

This paper looks at a specific aspect of the learning-to-teach problem, where the learner is assumed to have a teacher that selects training examples for the student according to a strategy. The teacher's strategy should also be  learned from data.  In this case the authors look at finding interpretable teaching strategies.  The authors define the "good" strategies as similar to intuitive strategies (based on human intuition about the structure of the domain) or strategies that are effective for teaching humans.
The suggested method follow an iterative process in which the student and teacher are interchangeably used. At each iteration the teacher generates  examples based on the students current concept.

I found it very difficult to follow the claims in the paper. Why is it assumed that human intuition is necessarily good?  The experiments do not answer these questions, but are designed to show that the suggested approach follows human intuition. There are not enough details to get a good grasp of the suggested method and the different choices for it,  and similarly the experiments are not described in a very convincing way. Specifically - the domains picked seem very contrived,  there actual results are not reported, the size of the data seems minimal so it's not clear what is actually learned.
How would you analyze the teaching strategy in realistic cases, where there is no simple intuitive strategy? This would be more convincing.

---

> ### Author Response · Authors · 2018-01-02
> **Purpose of our work**
>
> We wish to clarify that the goal of our work is to investigate the scientific question, “What would lead to human interpretable teaching strategies in neural networks?”. We focus on interpretability because it is, as pointed out by the other reviewers, important for a variety of problems [1,2,3]. The hypothesis that we aim to test in the paper is that training the student and teacher in an iterative fashion will lead to more interpretable strategies than training the student and teacher jointly.
>
> In order to evaluate “interpretability”, we need to operationalize it. Unfortunately, there is no consensus on what “interpretable” means [4,5], so instead, we devise two metrics that we hope together are robust in capturing what we mean by “interpretable”. The two metrics are (1) how close the teacher strategy is to human-designed strategies in each domain (2) how effective the strategies are for teaching humans.
>
> We believe that the reviewer’s statement “there actual results are not reported” is based on a misunderstanding of what our goal is. We do report on the above metrics, and based on these two metrics find support for our hypothesis.
>
> [1] NIPS Interpretable Machine Learning Symposium (2017) http://interpretable.ml/
> [2] DARPA Explainable AI Program https://www.darpa.mil/program/explainable-artificial-intelligence
> [3] ICML Tutorial on Interpretable Machine Learning (2017) http://people.csail.mit.edu/beenkim/icml_tutorial.html
> [3] Doshi-Velez, F., & Kim, B. (2017). Towards a rigorous science of interpretable machine learning.
> [4] Lipton, Z. C. (2016). The mythos of model interpretability.

---

### Decision · Program_Chairs · 2018-01-29
**ICLR 2018 Conference Acceptance Decision**

**Decision:**

Reject

**Comment:**

The paper proposes iterative training strategies for learning teacher and student models. They show how iterative training can lead to interpretable strategies over joint training on multiple datasets. All the reviewers felt the idea was interesting, although, one of the reviewers had concerns about the experimentation.

However, there is a BIG problem with this submission. The author names appear in the manuscript thus disregarding anonymity.